# A Review of Piezoelectric Footwear Energy Harvesters: Principles, Methods, and Applications

**DOI:** 10.3390/s23135841

**Published:** 2023-06-23

**Authors:** Bingqi Zhao, Feng Qian, Alexander Hatfield, Lei Zuo, Tian-Bing Xu

**Affiliations:** 1Department of Mechanical Engineering and Aerospace, Old Dominion University, Norfolk, VA 23529, USA; bzhao004@odu.edu (B.Z.); ahatf001@odu.edu (A.H.); 2Department of Mechanical Engineering Technology, The Behrend College, Pennsylvania State University, Erie, PA 16563, USA; 3Department of Naval Architecture and Marine Engineering, University of Michigan, Ann Arbor, MI 48109, USA; leizuo@umich.edu

**Keywords:** piezoelectric, energy harvesting, human walking, footwear, wearable, elastic energy, power, flextensional harvester

## Abstract

Over the last couple of decades, numerous piezoelectric footwear energy harvesters (PFEHs) have been reported in the literature. This paper reviews the principles, methods, and applications of PFEH technologies. First, the popular piezoelectric materials used and their properties for PEEHs are summarized. Then, the force interaction with the ground and dynamic energy distribution on the footprint as well as accelerations are analyzed and summarized to provide the baseline, constraints, potential, and limitations for PFEH design. Furthermore, the energy flow from human walking to the usable energy by the PFEHs and the methods to improve the energy conversion efficiency are presented. The energy flow is divided into four processing steps: (i) how to capture mechanical energy into a deformed footwear, (ii) how to transfer the elastic energy from a deformed shoes into piezoelectric material, (iii) how to convert elastic deformation energy of piezoelectric materials to electrical energy in the piezoelectric structure, and (iv) how to deliver the generated electric energy in piezoelectric structure to external resistive loads or electrical circuits. Moreover, the major PFEH structures and working mechanisms on how the PFEHs capture mechanical energy and convert to electrical energy from human walking are summarized. Those piezoelectric structures for capturing mechanical energy from human walking are also reviewed and classified into four categories: flat plate, curved, cantilever, and flextensional structures. The fundamentals of piezoelectric energy harvesters, the configurations and mechanisms of the PFEHs, as well as the generated power, etc., are discussed and compared. The advantages and disadvantages of typical PFEHs are addressed. The power outputs of PFEHs vary in ranging from nanowatts to tens of milliwatts. Finally, applications and future perspectives are summarized and discussed.

## 1. Introduction

Energy harvesting is defined as the conversion of the ambient energies present in the environment in various forms into usable electrical energy for powering electronic devices, sensors, and circuits [1]. This technology has been developed rapidly in recent years, driven by the fact that the burning of fossil fuels releases a large amount of carbon dioxide and greenhouse gases into the air, leading to climate changes and global warming [2]. Another driving force is the local power sources for wearable sensors, portable electronics, health monitoring systems, and wireless devices. As the clean energy revolution is taking place, including solar, wind, water, geothermal, bioenergy, and nuclear energy [3,4], research on new energy sources accelerates. Among these new energy sources, mechanical motion/vibration is one of the most investigated types due to its abundance, accessibility, and ubiquity in the environment [5,6]. Mechanical energy, including kinetic energy and potential energy, could be obtained from industrial machinery, automotive, human motion, large-scale buildings, and ocean waves [7]. Energy harvesters are considered to be promising distributed power sources for low-power portable electronics and wearable sensors [8,9]. Unlike conventional chemical batteries, which present issues relating to limited lifespan, environmental pollution, and recharging [10], energy harvesting is largely maintenance free and environmentally friendly [11].

Human mechanical energy and environmental mechanical energy are intensively exploited due to their abundance in daily life [12]. For example, the mechanical energy from human walking and running can be collected by energy harvesters assembled in shoes. The most common methods for mechanical-to-electric conversion mechanisms are piezoelectric [13,14,15,16,17,18,19], electromagnetic [20,21,22,23,24], triboelectric [25,26,27,28,29], and their hybrid derivatives [30,31,32,33], each with its own advantages and disadvantages, as illustrated in Table 1. In terms of efficiency, piezoelectric energy harvesting generally achieves good conversion efficiency in small volume space compared to electromagnetic systems, making it more suitable for low-power and low-profile applications where both energy and space are crucial [34,35,36]. Electromagnetic energy harvesting, while offering high power output, tends to require a large volume space [37]. Triboelectric energy harvesting can be scaled up or down, but its power output is generally low because of significantly high internal electrical impedance [12,38,39].

The choice between these approaches is application dependent, but the piezoelectric mechanism has been investigated predominantly, owing to the merits of its high energy density, high capacitance, low mechanical damping, easy shaping, and implementation [40]. Piezoelectric materials can generate electricity because the central symmetry of the crystal structure is broken under the action of the external force, forming a piezoelectric potential [7]. Among existing piezoelectric materials, lead zirconate titanate (PZT) and polyvinylidene fluoride (PVDF) are two of the most popular and cost-effective materials for energy harvesters mounted in shoes. Compared to PZT ceramics, PVDF has considerable flexibility, good stability, and is easy to handle and shape [41]. But PZT has the advantages of high mechanical-electric coupling factors, producing larger power, and easier integration with force amplification frames [40,42].

This paper gives a comprehensive review of the technology developments and research trends of piezoelectric footwear energy harvesters (PFEHs). The paper is organized as follows. The background and motivations for PFEHs research and developments are introduced in Section 1. The fundamentals of piezoelectric properties for footwear energy harvesters are briefly presented in Section 2. The force and dynamic energy distribution on the footprint, which includes the foot pressure, ground force reactions, and displacement, as well as acceleration during human walking, are reviewed in Section 3. Section 4 discusses the energy flow from human walking to the harvested energy through the PFEHs and the methods to improve the energy conversion efficiency. Following on the fundamental knowledge learned from Section 2, Section 3 and Section 4, the major PFEH structures and mechanisms on how the PFEHs capture mechanical energy from human walking to piezoelectric structures are summarized in Section 5. The main structures are classified into four types, including flat plate, curved, cantilever, and flextensional harvesters. The current applications and future perspectives of the PFEHs are presented in Section 6. Finally, Section 7 briefly summarizes this work.

## 2. Fundamentals of Piezoelectric Properties for Footwear Energy Harvesters

In 1880, Jacques and Pierre Curie discovered that certain crystals, such as quartz and tourmaline, create electrical charges when subjected to pressure; they called this phenomenon the “piezoelectric effect.” Later, it was discovered that piezoelectric materials could be deformed by electrical fields. This effect is known as the “inverse piezoelectric effect” [43]. Piezoelectricity is defined by Berlincourt [44] as the ability of a material to generate an internal electric field when subjected to mechanical stress or strain, while Erturk and Inman [45] defined it as a form of coupling between the mechanical and electrical behaviors of ceramics and crystals belonging to certain classes. Tension and compression generated voltages of opposite polarity, proportional to the force applied [46]. The basic relationships between the electrical and elastic properties of piezoelectric materials are given by
(1)$$\left[\begin{array}{c} D\\ S \end{array}\right]=\left[\begin{array}{cc} d & \varepsilon^{T}\\ s^{E} & d^{t} \end{array}\right]\left[\begin{array}{c} T\\ E \end{array}\right],$$
where D and E represent the electric displacement and electric field; S and T refer to strain and stress; d and $d^{t}$ are the matrices for piezoelectric charge coefficient and its transpose; $\varepsilon^{T}$ is the dielectric permittivity under a constant stress T; and $s^{E}$ is the elastic compliance under a constant electrical field E.

Due to the nature of human walking speed, most footwear energy harvesters work at a frequency of around 1 Hz, which is much lower than the resonant frequency of the piezoelectric elements of the devices, so the piezoelectric elements can be treated as parallel plate capacitors [8]. For a piezoelectric element with a surface area A and thickness t subjected to a stress σ, the total electric energy U can be roughly estimated by
(2)$$\begin{array}{cc} U=\frac{1}{2}QV & =\frac{1}{2}\left(d\times \sigma \times A\right)\left(g\times \sigma \times t\right)\\ & =\frac{1}{2}d\times g\times \sigma^{2}\times Volume \end{array}$$
where Q and V are the electric charge and voltage on the electrodes. The charge coefficient d and the voltage coefficient g correspond to the stress and electric field directions. Equation (2) shows that for a high-power density of the piezoelectric element, the d×g value should be high.

There are around two hundred piezoelectric materials used in different areas [47], including single crystal, lead-based piezoceramics, lead-free piezoceramics, and piezopolymers. Maamet et al. [5] summarized the main characteristics of piezoelectric materials, as shown in Table 2.

Table 3 summarizes the properties of typical piezoelectric materials. Single-crystal materials and natural single-crystal materials such as quartz have very poor crystal stability and a limited degree of freedom [48]. Compared with other materials, ZnO has a weaker piezoelectric coefficient. Although relaxor piezoelectric single crystals, such as PMN-PT and PZN-PT, have the highest piezoelectric constants, they are not popularly selected for energy harvesters because they are very expensive, and they are more sensitive to the environment temperature due to low phase transition temperatures [49]. Lead zirconate titanate (PZT) is the most common lead-based piezoceramic material for energy harvesting [50]. Despite its toxicity due to the presence of lead, it has high piezoelectric constants, low dielectric loss, and low manufacture cost. In addition, PZT multilayer stacks also has the advantage of high large load capability in it’s length direction. Lead-free piezoceramics, such as BaTiO_3_, usually have a lower transduction efficiency. They are also more expensive than the PZT [51]. Piezoelectric polymers are a great candidate for piezoelectric energy harvesting applications due to their low density and soft elasticity. These polymers also generate suitable voltage with sufficient power output, despite their low power density, and they can resist high driving fields because they have a high dielectric breakdown and possess maximum functional field strength. Furthermore, they have a low fabrication cost, and the processing is faster compared to ceramic-based composites [52,53,54,55]. Therefore, PZT and PVDF are two of the most popular and cost-effective materials for energy harvesters.

Generally, piezoelectric materials have two main strain modes, “$d_{31}$” mode and “$d_{33}$” mode. The first subscript number “3” denotes the poling direction, which is the same as the output voltage direction, while the second number is an indication of the direction of the applied force. As shown in Figure 1, the “$d_{31}$” mode means that the force direction is perpendicular to poling voltage direction. In contrast, the force direction is parallel to poling voltage direction in “$d_{33}$” mode. For the same piezoelectric material, the “$d_{33}$” is usually greater than or equal to two times of “$d_{31}$”. More importantly, the mechanical-to-electrical energy conversion efficiency of the “$d_{33}$” mode is 3~5 times [62,63] higher than the “$d_{31}$” mode. However, the “$d_{31}$” mode is more popular than the “$d_{33}$” mode in energy harvesting applications because (i) “$d_{31}$” mode can be simply applied to cantilever beam-type piezoelectric harvesters, and (ii) “$d_{33}$” mode needs more advanced piezoelectric harvester structure configurations, such as flextensional harvesters [14,40,63,64,65,66,67] and multistage amplification harvesters [68,69,70].

## 3. Force and Dynamic Energy Distributions on Foot Print

Human walking offers sufficiently harvestable, convertible, and continuous energy sources. In particular, the foot motion could produce both acceleration and large force excitations due to leg swing and heel strike. The foot structures produce mechanical work through elastic (e.g., Achilles tendon, plantar fascia) or viscoelastic (e.g., heel pad) mechanisms, or by active muscle contractions [71]. Research has shown that the foot itself behaves as a spring damper that stores and returns mechanical energy, providing considerable metabolic energy saving during human walking and running [72]. The passive elastic tissues inside human feet substantially play the role of spring-like structures, and other tissues dissipate energy as dampers in mechanical systems. Foot pressure and the large ground reaction force created during human walking are direct kinetic energy sources that can be harvested by using piezoelectric transducers. The leg swing and heel strike could lead to accelerated motion and inertia under the foot, which could be harvested by using both piezoelectric and electromagnetic transducers. Both the magnitudes and durations of the ground reaction force (GRF), as well as accelerations under the foot, are walking speed dependent. Therefore, it is essential to fully understand the plantar pressure, ground reaction forces, and acceleration during human walking, jogging, and running for the design of footwear power generators and for the choice of energy transduction mechanism. This section examines the harvestable energy sources under a foot in the form of foot pressure, ground reaction forces, and acceleration.

### 3.1. Foot Pressure

Studies on foot pressure were initially driven by the medical field to understand foot deformities and foot illness. Foot pressure contains valuable information regarding human foot morphology that differs by gender, age, body weight, and healthiness [73]. The measurement of foot pressure can be used as an indicator of diseases and abnormalities because foot pressure varies with respect to the subject’s health status, age, and activities. With the development of technology, extensive studies have been conducted on foot pressure on a qualitative and quantitative basis to understand information hidden beneath the foot in interaction with the surface in contact, for ergonomic, sports, clinical diagnosis, and human gait, and posture evaluation. From the perspective of footwear energy harvesting, measurement of pressures at the foot–shoe interface could provide more sophisticated information for the design of wearable insole energy harvesters. Identifying the area of maximum pressure during walking, jogging, and running could not only locate the optimum location of insole-type energy harvesters but also define the design constraints and conditions.

Foot pressure is usually measured by two kinds of plantar pressure measuring systems now commercially available on the market: pressure platforms (force plate) and in-shoe systems (insole) [74]. Force transducer cells of different types, such as capacitive sensors, piezoelectric elements, or strain gauges, are embedded in force plates and insole devices to measure the pressure under the foot. Pressure insoles generally provide reliable force and plantar pressure data, but the impact and propulsive force measurements were significantly less in magnitude than those measured with a force plate [75]. To quantify the distribution of pressure, the foot is typically divided into different regions, also referred to as a mask. Among them, the division of the foot into 10 regions, which, as shown in Figure 2a, includes the heel, midfoot, first, second, third, fourth, and fifth metatarsal heads, hallux, second toe, and third to fifth toes has been widely used. Figure 2b shows the distribution of maximum foot pressure for one step measured using shoe insoles with 99 capacitive sensors [74], where the maximum pressure is located at the heel and hallux. This research also concluded that aging affects the dynamics of foot pressure distribution during normal walking, and elderly people show lower pressure at the calcaneus and hallux regions compared with young people. An increase in body mass shows a positive relationship between the peak and mean plantar pressure variables for most plantar regions [76]. In addition to age and body mass, footwear was also reported to contribute to the change in plantar pressure. For example, Wiegerinck’s study on the plantar pressure of 37 athletes at a self-selected running speed shows that the total foot peak pressure in the racing flat shoes was 446.6 ± 77.25 kPa, while it was 407.3 ± 91.7 kPa in the training shoes [77].

The distribution of mean plantar pressure reported in the literature over the 10 regions is summarized in this section and tabulated in Table 4. Despite the discrepancies in the distribution of pressure resulting from different measurement systems, test subjects, and experiment setups, the maximum plantar pressure was found under the heel, hallux, and then the first to fifth metatarsal heads. This finding provides a sound basis for locating insole energy harvesters that are designed directly to harvest energy from normal force input under the foot, such as piezoelectric discs and stack-based energy harvesters. For these types of harvesters, large plantar pressure could usually result in a higher power output.

The kinetic energy under the foot during human walking is transmitted to the midsole consisting of a layer of elastic material between the foot and the ground. The energy is then partially stored as elastic strain energy, recovered back to the foot, and dissipated as heat inside the midsole. Shorten [84] studied the energy exchange and the spatial distribution of work and energy changes in the midsole during a running step. It was reported that a total of 11.5 J of work was performed on the midsole by a male subject of 76.0 kg at the running speed of 3.8 m/s, of which 7.9 J was recovered back to the foot (work performed by the midsole), and 3.6 J was dissipated as heat. The effects of running speeds on the energy stored and dissipated by the midsole were also examined, and the results are reported in Figure 3.

### 3.2. Ground Reaction Forces (GRF)

The ground reaction force (GRF) is one of the most common biomechanical parameters in gait, which includes both the magnitude and direction of loading applied to the foot during walking. During human locomotion, the GRF from the ground provides for propulsion and equilibrium control. The GRF is usually broken down into its three orthogonal components—vertical, anteroposterior, and mediolateral forces, respectively, among which the vertical component is dominant and easiest to quantify and is of most interest in energy harvesting. For example, for a piezoelectric stack-based footwear energy harvester, the generated voltage is proportionally correlated to the input force. This fact makes the large ground reaction force at the heel highly desirable for piezoelectric stack-based energy harvesting. The vertical ground reaction forces exceed horizontal forces by a factor of five or more, and the former exceed lateral forces by greater margins [85], which makes it the main excitation source for kinetic energy harvesting from human walking. The vertical ground reaction force was reported to range from 1.1 to 1.3 times body weight (BW) depending upon walking speed [86].

Data collected on twenty adult males during a running stance show that the average vertical GRF increased significantly from 1.40 BW at 3.0 m/s to 1.70 BW at 5.0 m/s [87]. At a moderate pace of 3 m/s, for runners who land on their rear foot, the vertical component of the GRF quickly rises and falls, forming the impact peak (1.6 BW). The GRF data collected from 20 adult males by Munro et al. [87] suggested that the maximum impact force increased in a linear manner from 1.6 BW to 2.3 BW over the examined speed range of 3–5 m/s. Milner et al. [88,89] reported that during running, the vertical forces placed on the body range is from 2.5BW to 2.8BW. Keller et al. [90] found similar trends for 23 subjects of recreational athletes, including 13 males and 10 females, in that that the vertical GRF increased linearly during walking and running from 1.2 BW to approximately 2.5 BW at 6.0 m/s. The impact force at a heel induced by heel strike during walking and running produces substantial mechanical energy usually absorbed and damped by the heel pad and passive tissues. Chi and Schmitt [91] estimated the amount of mechanical energy a heel pad has to absorb during impact loading of walking and running and found the foot is subject to total energy ranging from 0.24 to 3.99 J before each heel-strike, and the impact force for walking and running are 0.79 and 1.32 BW.

The vertical GRF shows different patterns at various walking and running speeds. The basic pattern of the vertical GRF during human walking has been extensively studied during the first half of the last century, Figure 4 graphically illustrates the typical pattern of the vertical GRF during human waking, which exhibits double peaks with an interjacent trough. The vertical GRF initially rises quickly and then falls, forming the first impact peak Fz1, which is about 1.6 BW at around 15–25% of the stance. The first peak vertical GRF is also referred to as the trust maximum force. It slowly decreases to the minimum Fz2 at the middle of the stance, and then it increases to a second peak which is around 2.5 BW, termed the maximum propulsive force, before decreasing prior to toe-off [92]. The vertical ground reaction force generally has a larger peak during the propulsive phase of the gain cycle (Fz2) than during the impact phase (Fz1). The vertical impact peak force during short-term downhill running is higher than the one during level running [93]. Keller et al. [90] also reported that the vertical GRF increased linearly during walking and running but remained constant at higher speeds. The vertical GRF–time histories only consisted of a single peak located at about 40–50% of the total stance time at higher running speeds for both female and male subjects, as shown in Figure 5, which is quite different from the double-peak pattern during walking. The mean vertical GRF ranged from 1.2 BW at 1.5 m/s to 2.5 BW. The greatest vertical GRF was found in the range of 2.5–3.0 m/s, which was recognized as the speed transition region between walking and running, and there were no significant increases observed at a speed over 3.5 m/s.

A rough calculation shows that the vertical ground reaction could create a maximum power of 2 W under the foot if the GRF is 1.2 times the body weight of 80 kg and there is a vertical displacement of 4 mm in the sole [94]. A more aggressive approximation indicates a 68 kg man walking at 3.5 mph, or 2 steps per second, could lead to a maximum power of 67 W by simply assuming a 5 cm vertical displacement at the heel [41]. The energy generated by the vertical GRF under the foot depends on different factors including the body weight, walking speed, and material of the sole. The numerical simulation and experimental measurement show that the generated mechanical power at the heel is only around 0.2 W for a male subject with a body weight of 84 kg and a walking speed of 4.8 km/h (1.3 m/s) when wearing a piezoelectric energy harvesting boot [40]. The resultant mechanical-to-electrical energy conversion efficiency of the footwear piezoelectric energy harvester is 4.7%.

### 3.3. Acceleration

In addition to plantar pressure and ground reaction force, acceleration under the foot is also a harvestable kinetic energy source. The acceleration under the foot is typically induced from heel strike and leg swing, and is utilized as the base excitation of energy harvesters, such as cantilever piezoelectric energy harvesters. The acceleration measured at various locations in the musculoskeletal system, such as the tibial and calcaneus, can be used to quantify heel strike-created impulsive loads. Research on 12 male subjects with either neutrally aligned or two pes planus feet has shown that the average peak-to-peak acceleration can be up to 6.75 g (±3.89 g), and there is no difference between the foot types [95]. Figure 6 shows a typical acceleration measurement for one complete gait cycle (from heel strike to heel strike) measured by an accelerometer placed at the calcaneus [95]. The heel strike creates large impact loads and high acceleration peaks at the heel, indicated by A and G in Figure 6, which are up to 7 g. Similar results were also reported by Eskofier et al. [96], where acceleration was used to classify different types of foot strikes including forefoot, midfoot, and rearfoot strikes. The repeated impact between the foot and the ground surface during walking and running provides repetitive acceleration excitation in energy harvesting. For example, the human body experiences approximately 3000 impacts with the ground during a 5 km run [97]. Small oscillations of the acceleration could be observed at B after the heel strike. There is hardly any motion at C, which was taken as the baseline. The small acceleration peak at D is attributed to the shock induced by the heel strike of the other foot, which was transmitted to the current foot through the musculoskeletal system, while the peak E is due to the swing phase of the instrumented leg.

Compared with the heel-strike-induced impact shock acceleration, the acceleration due to leg swing has a smaller amplitude and lower frequency components. Research has shown that the power spectral density (PSD) of the acceleration signals measured at the shank contains two major regions. The lower-frequency region of 4–6 Hz corresponds to the leg swing motion while the high-frequency region of 12–20 Hz is associated with the shock wave of the heel strike during the foot–ground impact [97,98]. Moro and Benasciutti [13] measured the acceleration at the heel by mounting an accelerometer close to the heel of a sports shoe, and quantitatively illustrated the vertical acceleration, velocity, and displacement during a complete gain cycle Figure 7a. A piezoelectric cantilever beam energy harvester with a proof mass was designed and placed inside the heel to convert the acceleration excitation into electricity. Figure 7b depicts the measured acceleration and voltage response of the piezoelectric energy harvester, and the average power of 395 μW was obtained from numerical simulations with optimal dimensions and electrical resistive load [13].

## 4. Energy Flow Chart

The energy flow chart of piezoelectric footwear energy harvesters, as shown in Figure 8, gives a better understanding of the principles of footwear energy harvesting and provides a guideline for creating and designing high-efficiency footwear energy generators. Liang and Liao firstly introduced the energy flow chart for piezoelectric energy harvesters [99]. Uchino divided the energy flow of piezoelectric energy harvesters into three phases [100,101], and Xu provides a better explanation [42]: Phase I is the mechanical energy capture and transportation processing; Phase II is mechanical-to-electrical energy conversion processing; Phase III is the electrical energy transportation to the outside of the harvester. Shabara et al. modified the diagram of energy flow [102]. Normally, piezoelectric footwear energy harvesting is performed via off-resonance dynamic processing. To enhance the comprehensibility of the energy flow associated with PEEHs, this paper divides the energy flow into four steps.

Step I is the mechanical energy capturing processing. First, the foot strike or leg swing energy is captured into deformable shoes/frames of PFEHs as elastic energy during walking or running. To increase the energy capture capability, the mechanical impedance matching between human foot/leg and the PEEH is the key considers for PEEH design. There is mechanical energy dissipation due to factors such as mechanical impedance mismatch as well as mechanical damping. The mechanical impedance of the material is defined as Z = (*ρ*c)^1/2^, where *ρ* is the density, and c is the elastic stiffness of the footwear/shoes.

Step II is the mechanical transferring process. In this step, the elastic deformation energy of the shoes or frames is transferred into the piezoelectric elements of PFEHs as either stretched or compressed elastic energy of the piezoelectric material. This step is only related to mechanical-to-mechanical transportation. It should be noted that not all the mechanical energy is transferred from the foot to the piezoelectrical materials. There is mechanical energy dissipation due to factors such as mechanical impedance mismatch, as discussed in step I, as well as mechanical damping. Uchino [101] suggested that the receiving part of the mechanical energy in the piezo devices should be designed to match the mechanical impedance of the vibration source to have a higher energy transferring rate. In addition, part of the mechanical energy goes back from the piezoelectric material to the shoes and then to the foot, which can be seen as a reaction power to support walking or running. Various mechanics, such as curved piezoelectric structures and flextensional harvesters, are applied to capture mechanical energy into piezoelectric structures [14,69,103,104,105,106,107].

Step III is mechanical-electrical energy conversion processing. Once the piezoelectric material is deformed, a surface charge with electrical potential (voltage) is generated in response to applied mechanical stress. In this process, the elastic potential energy of the piezoelectrical material is converted into electrical energy. The mechanical-to-electrical energy conversion rate (efficiency) can be evaluated by the square electromechanical factor $k_{ij}^{2}.$ Ceramic suppliers’ specifications usually provide mechanical coupling factor values $k_{ij}$, which is a measure of the efficiency of energy conversion in a piezoelectric ceramic. While a higher $k_{ij}$ value is generally preferred for efficient energy conversion, it does not take into account losses due to dielectric or mechanical factors, or the possibility of unconverted energy recovery [108]. Several approaches for increasing energy conversion efficiency by using “$d_{33}$” mode piezoelectric ceramic structures and “$d_{31}$” mode PVDF piezoelectric structures were researched in previous studies [14,40,66,69].

Step IV is the electrical-electrical energy transferring process. In this process, the generated electrical energy in the piezoelectric structures can be (i) directly applied to an electrical load and (ii) stored in an energy storage unit, such as a battery or a supercapacitor, for future use as a renewable power source. The energy delivery and storage from a piezoelectric structure can be found in Xu’s articles [42]. For energy storage from a piezoelectric energy harvester, an AC/DC voltage converter (electrical rectifier) is needed to convert the generated AC voltage to DC voltage before charging a battery or a supercapacitor. Energy storage from a piezoelectric structure to a battery/supercapacitor is a complex process which is not well addressed yet. The electrical energy loss in this process includes reflection by electrical impedance mismatching, voltage mismatching, voltage drop in diodes, and unusable resistive loads. To increase energy transport efficiency, impedance matching by optimized circuit design and voltage level optimization by applying a transformer (DC/DC converter) are used [101].

In summary, the energy flow of piezoelectric energy harvesters is a crucial aspect of the guideline for PFEH designs. The key issues for a PFEH design include capturing more mechanical energy into the elastic deformation energy of shoes, increasing energy transportation efficiency to piezoelectric structures, increasing mechanical-to-electrical energy efficiency by piezoelectric structure deformation orientation selection, and increasing electrical energy transfer efficiency by electrical circuit design.

## 5. Structures and Configurations of the Piezoelectric Footwear Power Generators

Various piezoelectric footwear power generators have been developed over the past two decades to convert mechanical energy under the foot to usable electricity. To fit in the limited space between the foot and the ground, the structures of piezoelectric transducers were designed diversely. A majority of the designs are for directly harvesting the plantar foot pressure and ground reaction forces, such as the piezoelectric insole and sole energy harvesters [109,110,111]. Because shoe soles bend during walking, the insole and sole piezoelectric energy harvesters are usually designed with flexible structures and soft piezoelectric materials, such as PVDF. Stiffer structures could lead to discomfort and invasiveness, or even changes the gait pattern during human walking. The locations of piezoelectric transducers inside shoes mainly depend on the energy sources targeted to harvest. While insole harvesters are designed to harvest foot pressure, piezoelectric stacks and thunders are particularly favored for scavenging ground reaction forces and are therefore usually placed in the heel and forefoot [13,14,16,112]. To improve the power generation performance, piezoelectric transducers with force amplifiers are developed to amplify the ground reaction force. Transducers for harvesting large ground reaction forces and pressure under the foot need capacity and durability to withstand the repetitive impact forces to prevent early fatigue failure. Generally, the structures of piezoelectric transducers for foot wearable energy harvesters in the literature have been categorized into four types: flat plate, curved shape, cantilever, and flextensional design. Piezoelectric cantilever harvesters are more suitable for harvesting the acceleration generated by either the heel strike or leg swing. The positions where piezoelectric transducers are implanted vary from heel, insole to the sole. This section discusses the state-of-the-art research on piezoelectric footwear energy harvesters in terms of their structures and locations.

### 5.1. Flat Plate

A flat plate footwear power generator usually has a simple structure. As illustrated in Figure 9a,b, it consists of thin piezoceramic disks or PVDF foils, coated with metal electrodes on both sides. The piezoceramic disks or PVDF films are attached to a metal shim fixed on the edges of the clamping ring or a plastic core, which act as a passive layer to protect the thin piezoelectric films and make the piezoelectric material stretch when the flat plate is subjected to a compressive force [113,114]. The flat plate energy harvester mainly uses the “$d_{31}$” mode for energy harvesting. When the force from the foot is applied on the up surface, the thin films will deform and then stretch horizontally. It will have a negative charge on one face and a positive charge on the opposing face, and then it will generate the voltage on the surface [115,116,117]. Because the shape of the foils is plain, there is not too much deformation when it is compressed, and the energy conversion rate from mechanical energy to electrical power is low. To obtain a higher power output, several piezoelectric films will be used in this type of structure, connected either in series [118] for high output voltage or in parallel [119] for high output current.

The flat plate energy harvesters are mostly thin and flexible [19,103,120,121,122,123,124], so they are usually attached to the insoles, either mounted on them or embedded into them. Because the insoles contact the foot (or socks) directly, the force from the foot will strike the entire harvester device forthright, which can increase the output energy [27,125,126,127]. The material of the generator should be soft and flexible. Also, the generators need to be well-sealed, to protect them from sweat, dirt, and bacteria from the foot [128].

In 2001, Nathan S. Shenck et al. [103] from MIT explored two main methods, illustrated in Figure 10a, of “$d_{31}$” mode piezoelectric shoe energy harvesters. One of them was a PVDF stave with an elongated hexagon shape, and it was put under the ball of the foot. As shown in Figure 10b, it had a 2 mm flexible plastic substrate, atop and below which were epoxy-bonded multi-PVDF layers. The PVDF sheets were connected in parallel to lower the impedance and increase the current. It worked in “$d_{31}$” mode. When the stave was subjected to the force from the foot, the PVDFs on the outside surface were pulled into expansion, while those on the inside surface were pressed into contraction. The PVDF staves produce ±60 Volts peak voltage and 1.1 mW average power at a walking frequency of 1 Hz.

Ahmad et al. [19] designed a footwear power generator with five piezoceramic discs on the insole. Figure 11a illustrates two discs placed at the ball area and three placed at the heel as both locations have a high impact force during walking. The five discs were connected in a series. The peak voltage during the walking was 14.1 Volts, and the output power was 1.41 mW. Similarly, Parul and Puneet [129] put two PZT buzzers inside the insole at the ball and heel areas, as illustrated in Figure 11b. They investigated the effect of the positions and dimensions of the PZT buzzers on the power output. The experimental maximum power obtained was 0.2 mW. In 2016, Snehalika and Bhasker [124] designed a piezo generator with multi-PVDF films, as pictured in Figure 11c. The PFEH consisted of six layers of PVDF films connected in a series. Each layer contained nine small pieces of PVDF connected in parallel.

### 5.2. Curved Structures

Another type of PFEHs is the curved structure. Compared to the flat plate, the curved structure is usually more efficient because it can produce larger strain and capture more mechanical energy into piezoelectric structures with the arc-shaped force application mechanism [111]. Figure 12a shows the schematic of a typical curved structure, which consists of curved thin flexible piezoelectric films, such as unimorph PZT composite strips and PVDF. The foils are attached onto a metal or plastic core substrate which has high stiffness. The substrate plays two important roles. Firstly, it acts as a passive layer to be effectively subjected to the vertical force to the piezoelectric layer. More importantly, it turns the deformed piezoelectric layer into its original shape when the driving force is removed [105].

The curved structure intensifies the strain of the piezoelectric films when the force from the foot is applied on it. Based on our further understanding the mechanism of the references [102,111], Figure 12b shows the force transmitted in the curved structure. Assuming the force from the foot is $F_{foot}$, when the curved energy harvester is pushed down, the piezoelectric sheets on the up surface of the core substrate will contract, while those below the substrate will expand. The stretch/compression force of the piezoelectric films is amplified by the curved structure so that there will be more power output compared to the flat plate structures.

Early in 1997, Hellbaum et al. [104] invented a curved PZT unimorph “Thunder”, as shown in Figure 13, at NASA Langley. Thunder is the acronym of “Thin Layer Composite Unimorph Ferroelectric Driver and Sensor”, which is a unimorph strip of spring steel bonded to a flexible PZT composite. This “Thunder” unimorph can be pressed flat, but the reverse bend will make it crack. Shenck and Paradiso [103] used this structure to capture energy from walking. As illustrated in Figure 10a, a PZT “dimorph”, which consisted of two back-to-back, single-sided unimorph, was mounted under the heel-strike center. The two PZT unimorphs were connected in parallel. With a walking pace of 0.9 Hz, the dimorph produced an average voltage of 44 Volts and an average power of 8.4 mW.

To generate a high output power from the piezoelectric generator for wearable applications, Jung et al. [105] demonstrated a curved fusiform structure. The structure diagram is shown in Figure 14b. The structure consisted of two curved piezoelectric generators connected on the edges. Each of them comprised a curved plastic core substrate and two PVDFs attached on each side of the substrate surface. The gold layers served as electrodes. When the generator was attached on the insole, during 0.5 Hz frequency walking, around 25 Volts of average voltage and about 20 µA of average current were obtained. An alternative way of the curved type is to make multilayer piezoelectric films sandwiched between two wavy surfaces, as shown in Figure 15 [106]. The wavy shape structure was specially designed to enable the PFDV films to obtain a large longitudinal deformation, as well as to reduce the harvester’s thickness for the shoes’ limited inner space. An average output power of 1 mW was harvested during a walk at a frequency of 1 Hz.

### 5.3. Cantilever

The structure of the cantilever piezoelectric energy harvester is relatively simple. As shown in Figure 16, the beam is composed of two layers of piezoelectric films, with the left end fixed on a base and the right end free. The configuration is known as “unimorph” if there is only one piezoelectric layer bonded to the metallic layer, and “bimorph” if there are two piezoelectric layers [12]. Usually, there is a proof mass on the free end to lower the resonance frequency of the harvester. It would produce a damped oscillation when the free end is stressed [130]. It has been found that the power output and the resonance frequency of a cantilever energy harvester is tuned by the proof mass [64,131]. More importantly, the fundamental bending mode of a cantilever is significantly lower than the other vibration modes of the piezoelectric element. Therefore, it is suitable for low-frequency conditions for low self-vibration frequency [132], such as in walking and running. The power output of this structure relies on the beam materials, the shape of the beam, the weight of the mass, driving frequency, etc. [133,134,135,136]. Hence, appropriate mechanical and electrical optimizations are necessary.

Moro and Benasciutti [13] designed a shoe-mounted piezoelectric cantilever harvester excited by the heel swing acceleration during walking. As shown in Figure 17a, a rectangular PZT-5A piezoelectric cantilever was mounted inside the heel using a stainless-steel clamp. A tip mass was fixed at the free end of the cantilever. The measured power per footstep was approximately 13.8 μW. Fan et al. [137] proposed a shoe-mounted nonlinear PFEH. As shown in Figure 17b, it comprises a piezoelectric cantilever beam, which was coupled to a ferromagnetic ball, and a crossbeam. The ball was placed in a sleeve that guided the travel of the ball. The magnetic mass beneath the crossbeam and the magnetic ball was used to drive the mass on the tip. The power generated by the prototype ranged from 0.03 mW to 0.35 mW when the walking speed varied from 2 km/h to 8 km/h. Xin et al. [130] equipped a PVDF cantilever beam inside a heel. As illustrated in Figure 17c, the harvester is made of a driving spring, a metal lever, and three PVDF films. The spring would drive the PVDF to oscillate when the heel hit the ground and lifted. The harvester could provide a maximum output power of 0.48 mW.

### 5.4. Flextensional

The flextensional structure is also a typical energy harvester, including cymbal transducers and stacks energy harvesters. The cymbal structure was first invented by Newnham and Xu [107] in 1994 for ultrasonic transducer applications. Uchino et al. at Penn State University successfully used cymbal structure [101,138,139,140,141] for off-resonance mode piezoelectric energy harvesters and achieved 7.8% energy efficiency, which is the highest recorded from reports in the literature. Flextensional structures with piezoelectric elements have been used for footwear power generators. As shown in Figure 18a, the cymbal structure comprises a piezoelectric disk sandwiched between two metal substrates.

Similarly, the “$d_{33}$” mode piezoelectric multilayer stack-based flextensional harvester is shown in Figure 18c. Piezoceramic films of hundreds of layers stacked together are usually used for this type of structure because it is stiff and has high piezoelectric coefficients. The main difference between these two structures is that in the stacked architecture, piezoelectric materials utilize the “$d_{33}$” mode, which offers 3~5 times higher mechanical-to-electrical energy conversion efficiency, while in the cymbal architecture, piezoelectric materials utilize the “$d_{31}$” mode. Piezoelectric stacks can usually operate under large force input but off-resonance, owing to their very large axial stiffness and high natural frequencies. Multilayered piezoelectric stacks show much higher equivalent piezoelectric constants and thus result in large power output compared with the “$d_{31}$” mode cymbal design.

The force-amplifying structures can be used to increase the power output of the flextensional energy harvester. The frames of the structure serve as a mechanical transformer with a force amplification effect to transfer the compressive load force $F_{foot}$ into tensile force $2F_{h}$ along the horizontal direction. The free-body diagrams of the cymbal structure and stacked flextensional structure are illustrated in Figure 18b,d. The overall horizontal force 2$F_{h}$ of the piezoelectric material can be approximately expressed by
(3)$$2F_{h}=\cot\theta F_{foot}$$
where θ is the angle between the frame and the horizontal line. The amplify factor is cot⁡θ. If θ is small, then the force from the foot is enlarged greatly, which increases the power output. Because of the high stiffness of cymbal and stacked flextensional structures, the natural frequency of piezoelectric stacks is usually greater than 1 kHz, which is higher than most natural vibrations in the environment. The majority of cymbal transducers and piezoelectric stacked flextensional harvesters work at off-resonance and can withstand larger mechanical loads.

Leinonen et al. [16] fabricated a piezoelectric cymbal harvester for energy harvesting from walking. As shown in Figure 19a, the prototype consisted of a Ø 35 mm PZT-5H disk, two concave endcaps, and two brass rings which were attached between the piezoelectric layer and the endcaps and served as electrodes. The energy harvester was placed on the sole in the heel area and tested at different forces. The maximum power output was about 800 µW at a walking frequency of 1 Hz. Kuang et al. proposed a sandwiched piezoelectric transducer, which is a “$d_{31}$” mode rectangular flextensional harvester [112]. As shown in Figure 20a, two metal endcaps were used to amplify the applied load force to the “$d_{31}$” mode PZT plate. The PZT plate was sandwiched between two metal substrates. The use of the substrates significantly reduced the stress concentration on the PZT, thus increasing the load capacity. When the transducer was fit into a sole of a boot, it generated an average power of 2.5 mW at a walking speed of 4.8 km/h. Qian et al. designed a piezoelectric footwear harvester for energy scavenging from human walking using the flextensional force amplification frame to transmit and amplify the vertical heel-strike force to the inner piezoelectric stack deployed in the horizontal direction [14,40]. As shown in Figure 21B, in the design two heel-shaped aluminum plates are employed to gather and transfer the dynamic force over the heel to the sandwiched force amplification frames. An optimal force amplification frame is obtained by parameter optimization to achieve a large force amplification factor and efficient energy transmission. Two harvesters, including eight and six piezoelectric stacks, respectively, were manufactured and integrated into the heel of a size 9 Belleville boot (26 cm in length). The maximum average power outputs (delivered to matched resistive load) of the harvester with eight stacks achieved 9 mW/shoe at the walking speed of 3.5 mph (5.6 km/h), while the harvester with six stacks produced 14 mW/shoe.

Later, Qian et al. designed a harvester comprising four units of two-stage force amplification piezoelectric transducers sandwiched between two metal plates [69], as shown in Figure 21A. A large power output was achieved because the force applied to the “$d_{33}$” mode piezoelectric stacks was amplified twice by the two-stage force amplification frames. The prototype was tested on an Instron machine (as shown in Figure 21B) under a dynamic force with different amplitudes ranging from 80 N to 500 N. The average power outputs of 23.9 mW and 11 mW were achieved under 500 N input force at 2 Hz and 1 Hz, separately. The maximum average power attained under 400 N input force and 3 Hz was up to 32 mW, as shown in Figure 21C.

Table 5 summarizes the performance of some high-performance piezoelectric footwear energy harvesters from the last two decades. For each PFEH, the table lists structures, bibliography references, material, average power output, and the location of the transducers. Overall, most power outputs of PFEHs are in the milliwatts range. Research is continuing to improve the efficiency and power density while exploring new applications of the PFEHs.

## 6. Applications

### 6.1. For Military Missions

On the modern battlefield, technology must compete for space. Soldiers today are sensor platforms—part of a mobile network of users and platforms [142]. During a 72 h mission, a US networked rifleman will usually carry 16 pounds of batteries for devices such as GPS, smartphones, imaging systems, and communications gear. Along with water and ammunition, power contributes significantly to the physical burden.

Footwear power generators provide an alternative source of energy for soldiers to power their electronic devices and charge their batteries. This can be achieved merely by warfighters’ movement during the mission [124]. The PFEHs help to reduce the number of needed batteries, in a way, reducing the weight, giving the soldier a speed of movement and shorter reaction time. More importantly, during a long-range mission in some austere environment, there is no power outlet. When the batteries run down, wearable energy harvesters can still work as a reliable energy resource for low-power consumption devices such as GPS and communication devices, which can save lives.

### 6.2. For Health Care and Monitoring

Nowadays, there are a huge number of real-time health monitoring devices on the market. These devices can be used to monitor the heart rate, blood pressure, blood glucose, and body temperature, as well as walking steps, time spent exercising and even burned calories, especially for people with medical conditions and athletes [30]. With the development of microchips, the consumption of these types of equipment can be ultra-low. Footwear power generators could replace traditional chemical batteries as they can be a reliable power source for portable devices and wireless sensors [33]. Specifically, when the wearers of footwear energy harvesters are walking or running, the generator powers up the wearable devices and at the same time charges a pre-installed battery. And when they stop moving and have a rest, the fully charged battery now is on duty and supplies the electrical energy to the wearable equipment.

### 6.3. For Other Applications

#### 6.3.1. Night Safety

The road workers or traffic police will have a higher risk of secondary accidents when they deal with different situations on the road at night or in bad weather conditions due to low visibility [17]. Even with reflective clothing, sometimes it is hard to see when the light reflection angle is not right or when it is far away from the cars. Energy harvesting shoes can be utilized to solve this problem. Just simply connect the power generator to an LED light bulb, which can be either mounted on the shoes or attached to the reflective jackets. Once a foot hits the ground, the LED will give off a bright light. Then, as the other footsteps hit the road, the LED will shine again. So, the LED can continuously flash when the workers or police officers walk. This can strongly enhance the safety of these people during the night, without extra cost to the battery or the recharging electric energy.

#### 6.3.2. Frostbite Protection

During the winter or in the cold areas, outdoor activities such as hiking, skiing, and mountain climbing, or outdoor work such as snow cleaning, repairs, and so on will make the feet very cold and can lead to frostbite, particularly at the toes [143]. Even with proper and highly effective insulation to keep the moisture outside and heat inside the shoes, or with double-layer socks, the freezing environment can still make the foot uncomfortable. Recently, a shoe-heating system based on piezoelectric energy harvesting has been developed to warm the feet [144]. The piezoelectric power generators would convert mechanical energy to electrical energy. Then, the generated electrical power is used to generate heat inside the shoes by connecting a heating device, preferably positioned around the toe area of the footwear. The shoes also contain a fan that accelerates the circulation of warm air inside the shoes.

#### 6.3.3. Hiking

Electronics such as a smartphone, GPS, a headlamp, and so on are essential items in many hikers’ backpacks. When they run out of power in the backcountry, it can be a real drag. Footwear power generators offer a solution. During long-distance hiking, the shoes can continuously output electrical energy to charge up the battery as the hikers are walking. Even though the power output of each step may be small, thousands of steps’ energy is accumulated. There would be considerable energy contained in the battery, which is enough to charge the electronic products.

#### 6.3.4. Extreme Environment

In extreme environments, such as in the middle of the rainforest, at the peak of Everest, or in the polar regions, a GPS device is necessary to give directions when going off-road. Conventional GPS will only survive several days without charging due to the limited battery power capacity. A combat boot with a passive GPS tracker, powered by an insole energy harvester, has been developed. The GPS is specially designed to minimize energy consumption. It can periodically ping geographical coordinates to a satellite and relay this information to a command center [145].

## 7. Summary

This paper conducts an extensive review of the piezoelectric footwear energy harvesters published in the past two decades. Firstly, the fundamentals of piezoelectric properties for footwear energy harvesters are briefly introduced. Secondly, the force and dynamic energy distribution on footprint are reviewed to provide a guideline for PFEH designs. Furthermore, the energy flow from the human walking to the usable energy through the PFEHs and the methods to improve the energy conversion efficiency are presented. The major PFEH structures and mechanisms for how the PFEHs capture mechanical energy from human walking to piezoelectric structures are summarized. Based on the most popular PFEHs developed in the last two decades, the main structures are classified into four types, including flat plate, curved, cantilever, and flextensional. A specific piezoelectric energy harvester’s power output varies greatly, ranging from nanowatts to milliwatts, depending on not only the structures but also the materials and the locations of the PFEHs, as well as the walking speed, frequency, etc. So far, the highest average electrical power of 23.9 mW was harvested. Finally, the applications based on self-powered PFEHs are shown to highlight their significant potential for accelerating the development of wearable sensors and electronics, and the potential applications of PFEHs are summarized and discussed.

Although piezoelectric footwear energy harvesters have been studied for decades, the achieved power performance is still considerably lower than the expectation. Even though some prototypes have been tested, their reliability, stability, and comfort for wearing have not been fully studied yet. The current level of performance of piezoelectric materials makes it rather difficult to create an energy harvester that can effectively replace batteries as a major power source. Continuous reductions in the energy consumption of electronic devices, improvement of the performance of piezoelectric materials, and the development of new energy harvesting mechanisms are likely to be essential for the future of piezoelectric energy harvesting research and implementations.

## Figures and Tables

**Figure 1 sensors-23-05841-f001:**
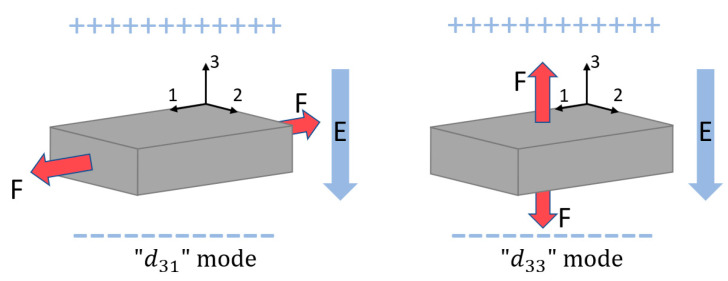
“$d_{31}$” mode and “$d_{33}$” mode piezoelectric material operations.

**Figure 2 sensors-23-05841-f002:**
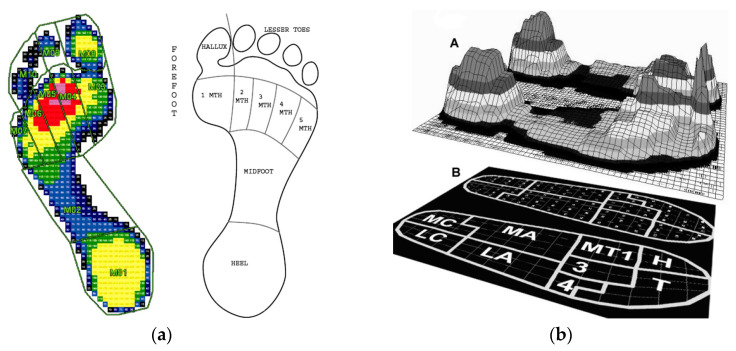
(**a**) The mask defining the 10 regions of the foot (the pressure from lower to higher: blue → green → yellow → red): left foot, Reprinted with permission from [78]. Copyright © 2023 Elsevier, right foot [79] Reprinted with permission from [79]. Copyright © 2023 Elsevier; (**b**) foot pressure distribution: (**A**) maximum pressure distribution, (**B**) the nine anatomical masks superimposed on the insole (MC = medial calcaneus, LC = lateral calcaneus, MA = medial arch, LA = lateral arch, MT1 = first metatarsal, 3 = second and third metatarsal, 4 = fourth and fifth metatarsal, H = hallux, and T = toes). Reprinted from [74] open resource.

**Figure 3 sensors-23-05841-f003:**
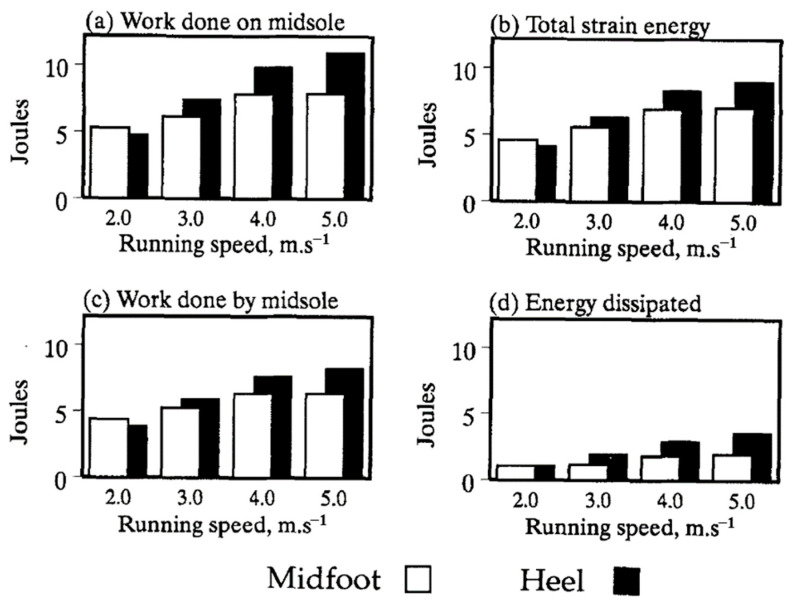
Effect of running speed on the energy stored and dissipated by the midsole [84]. Reprinted with permission from [84]. Copyright © 2023 Elsevier.

**Figure 4 sensors-23-05841-f004:**
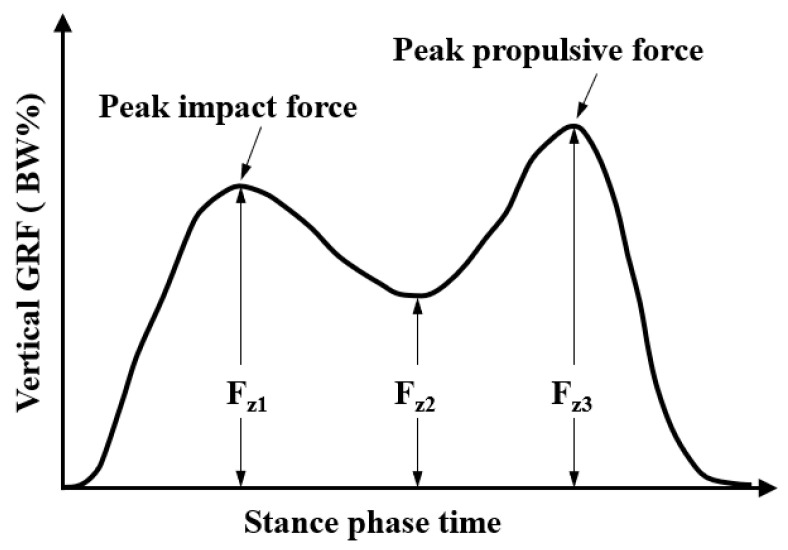
The typical pattern of the vertical ground reaction force below the foot in one gait during walking.

**Figure 5 sensors-23-05841-f005:**
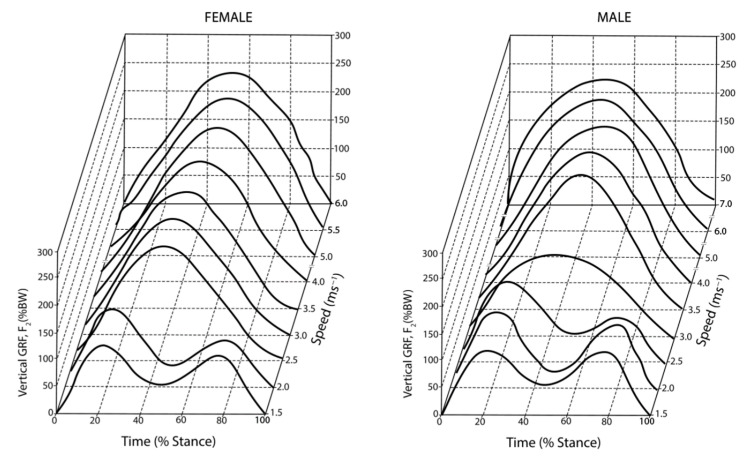
The vertical ground reaction force at different walking speeds [90]. Reprinted with permission from [90]. Copyright © 2023 Elsevier.

**Figure 6 sensors-23-05841-f006:**
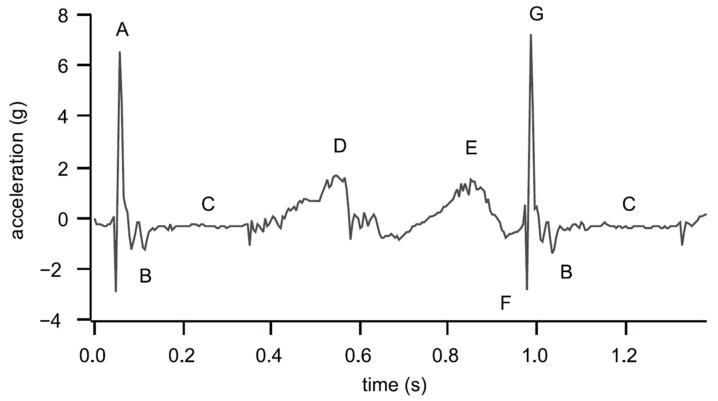
The acceleration measured at the calcaneus for one gait cycle: (A) previous heel strike, (B) oscillation after heel strike, (C) baseline, (D) heel strike of the opposite limb, (E) swing phase, (F) downward acceleration at heel strike and (G) upward acceleration at heel strike [95]. Reprinted with permission from [95]. Copyright © 2023 Elsevier.

**Figure 7 sensors-23-05841-f007:**
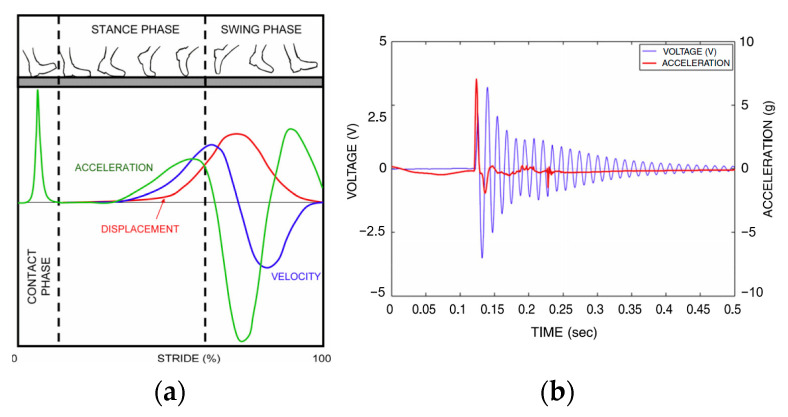
(**a**) qualitative relation between the acceleration at the heel and walking stance; (**b**) measured acceleration at the heel and voltage output of the piezoelectric energy harvester [13]. Reprinted with permission from [13]. Copyright © 2023 IOP.

**Figure 8 sensors-23-05841-f008:**
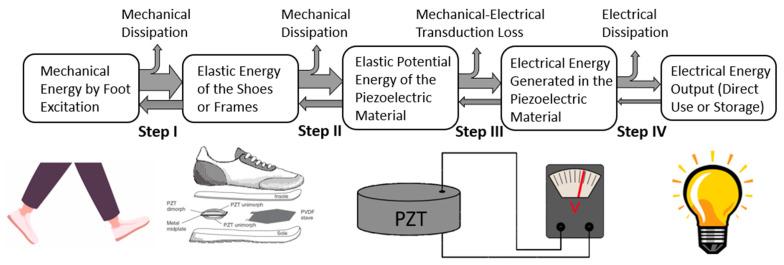
Energy flow chart of piezoelectric footwear energy harvesters [42,99,100,101,102,103]. Reprinted with permission from [103]. Copyright © 2023 IEEE.

**Figure 9 sensors-23-05841-f009:**
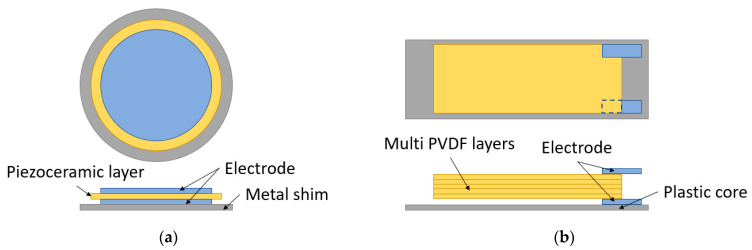
Flat plates for PFEHs: (**a**) a piezoelectric ceramic disc; (**b**) multi-PVDF films.

**Figure 10 sensors-23-05841-f010:**
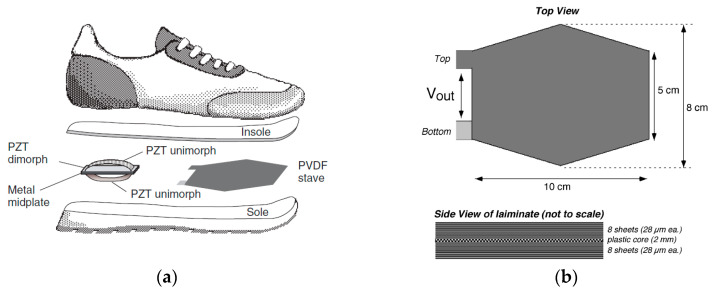
(**a**) Two approaches: a PVDF stave under the ball of the foot and a PZT dimorph under the heel [103] Reprinted with permission from [103]. Copyright © 2023 IEEE.; (**b**) layout of the PVDF stave [120]. Reprinted with permission from [120]. Copyright © 2023 IEEE.

**Figure 11 sensors-23-05841-f011:**
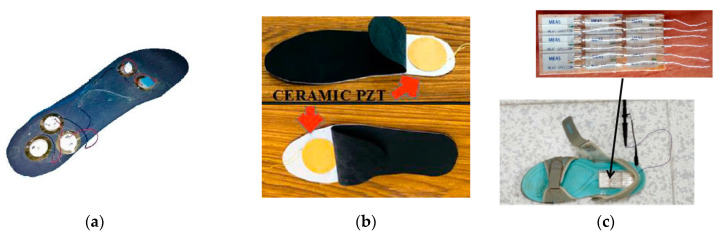
(**a**) Piezoceramic discs attached on the insole of ball and heel areas [19] Reprinted with permission from [19]. Copyright © 2023 IEEE.; (**b**) ceramic PET buzzers placed inside the insole [129] Reprinted with permission from [129]. Copyright © 2023 Springer.; (**c**) a shoe with six layers comprising nine PVDF films [124]. Reprinted with permission from [124]. Copyright © 2023 IEEE.

**Figure 12 sensors-23-05841-f012:**
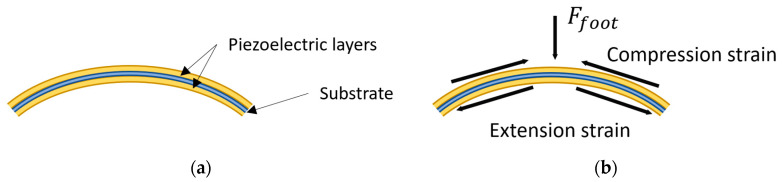
(**a**) A typical curved piezoelectric structure for PFEHs; (**b**) diagram of the force amplification for the curved piezoelectric structure. (Recreated and modified from reference [105]). Reprinted with permission from [105]. Copyright © 2023 Elsevier.

**Figure 13 sensors-23-05841-f013:**
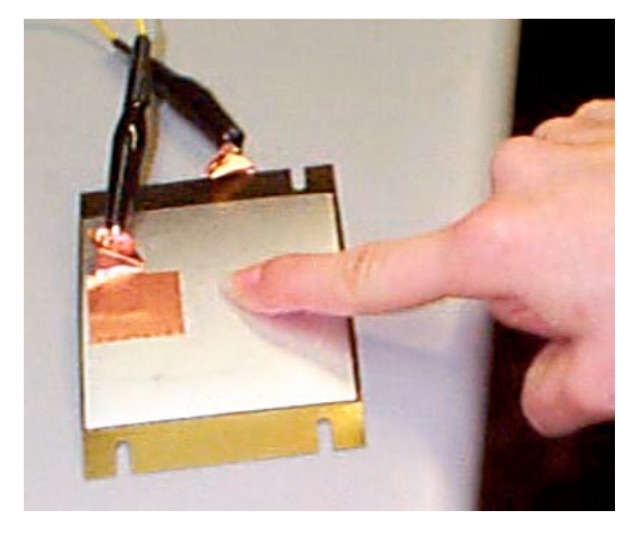
“Thunder” PZT unimorph [120]. Reprinted with permission from [120]. Copyright © 2023 IEEE.

**Figure 14 sensors-23-05841-f014:**
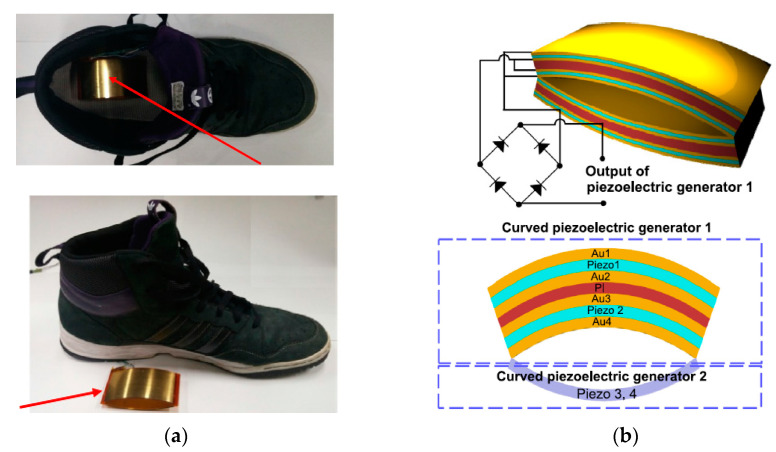
(**a**) A curved piezoelectric generator using PVDF; (**b**) structure diagram of the generator [105]. Reprinted with permission from [105]. Copyright © 2023 Elsevier.

**Figure 15 sensors-23-05841-f015:**
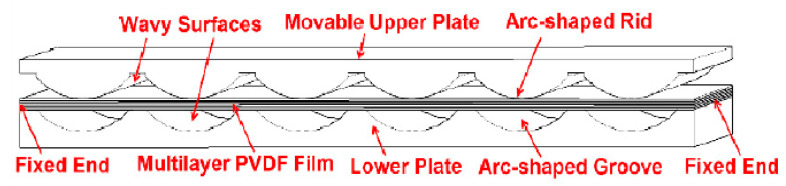
A wavy shape sandwich structure of the harvester [106]. Reprinted from open source [106].

**Figure 16 sensors-23-05841-f016:**
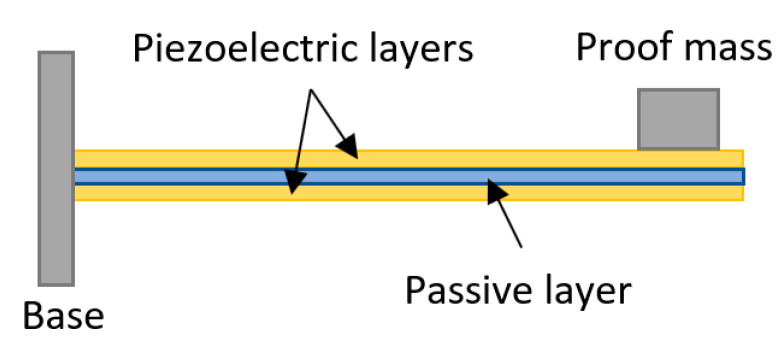
Piezoelectric cantilever bimorph.

**Figure 17 sensors-23-05841-f017:**
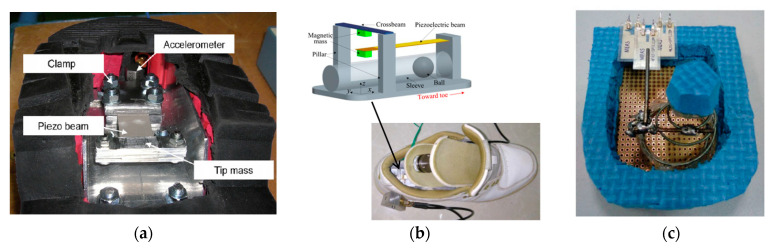
(**a**) Rectangular PZT-5A piezoelectric cantilever [13] Reprinted with permission from [13]. Copyright © 2023 IOP; (**b**) magnetically driven cantilever harvester [137] Reprinted with permission from [137]. Copyright © 2023 AIP Publishing; (**c**) PVDF cantilever harvester [130]. Reprinted with permission from [130]. Copyright © 2023 Taylor & Francis.

**Figure 18 sensors-23-05841-f018:**
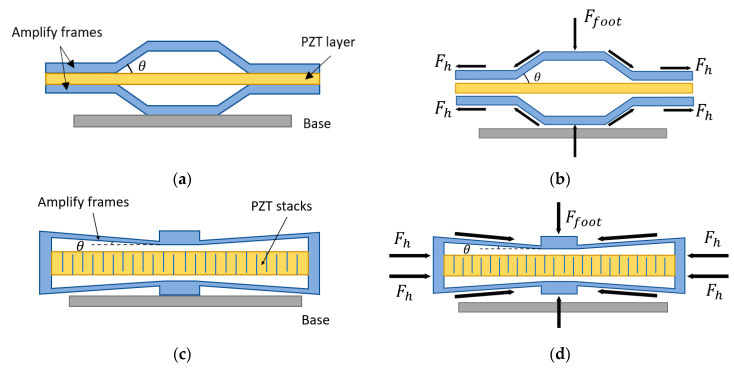
(**a**) Diagram of cymbal energy harvester; (**b**) diagram of the force reaction and amplification for the cymbal structure; (**c**) diagram of stacked flextensional energy harvester; (**d**) diagram of the force reaction and amplification for the stacked flextensional structure.

**Figure 19 sensors-23-05841-f019:**
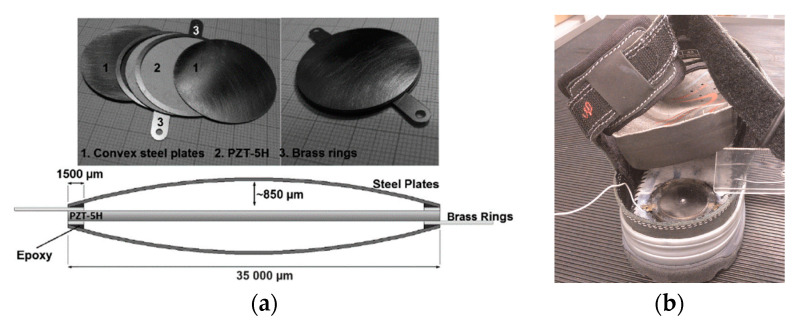
(**a**) The components; cross-section view of the prototype. (**b**) A circle cymbal harvester inside the sole of the shoe [16]. Reprinted with permission from [16]. Copyright © 2023 SAGE Publications.

**Figure 20 sensors-23-05841-f020:**
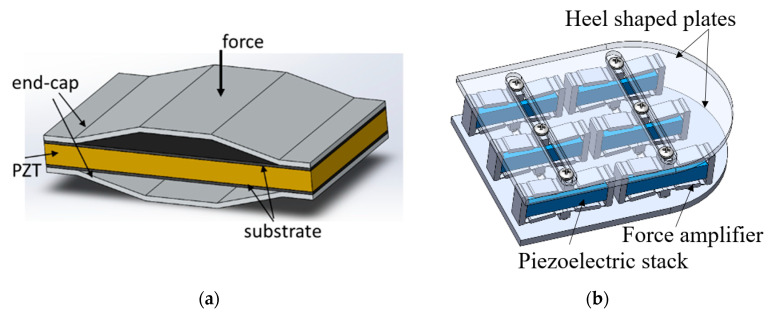
(**a**) A flextensional energy harvester with flex endcaps [112]. Reprinted from open source [112]; (**b**) the footwear energy harvester with flextensional force amplifier and piezoelectric stacks [40]. Reprinted with permission from [40]. Copyright © 2023 IOP.

**Figure 21 sensors-23-05841-f021:**
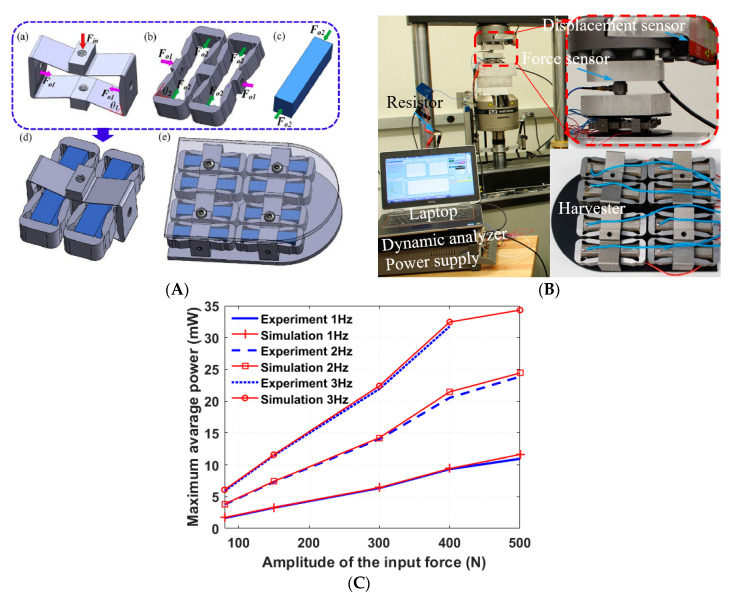
(**A**) A flextensional structure with two-stage force amplification: (a) second-stage FAF, (b) first-stage FAF, (c) piezoelectric stack, (d) two-stage piezoelectric transducer unit, (e) assembled two-stage PEH, (**B**) experimental setup; (**C**) maximum average power under different load levels and excitation frequencies [69]. Reprinted with permission from [69]. Copyright © 2023 Elsevier.

**Table 1 sensors-23-05841-t001:** Summary of Advantages and disadvantages of piezoelectric, electromagnetic, and triboelectric energy harvesting methods.

Energy Harvesting Method	Advantages	Disadvantages
Piezoelectric	High efficiency Small displacement and working space Compatibility with various vibrations	Fragility of some materials Limited power output
Electromagnetic	High power output Wide range of applications Long-term stability	Limited efficiency at small scale compared to other methods Specific environmental conditions may be required Bulkier setup
Triboelectric	Versatility in capturing energy from various sources Cost effective Potential for scalability	Large electrical impedance Limited power output Environmental considerations Efficiency challenges

**Table 2 sensors-23-05841-t002:** Characteristics of piezoelectric materials * [5].

Type	Description and Characteristics	Existing Solutions and Examples
Single-Crystal Materials	Monocrystals vertically grown on a substrate via the Bridgeman method or Flux method, etc.; Outstanding piezoelectric properties and are mostly used for sensors and actuators; Depending on the growing technique, they can have different nanostructure forms.	Zinc-Oxide (ZnO); Lead Magnesium Niobate (or PMN)-based nanostructures: PMN-PT.
Lead-based Piezoceramics	Polycrystalline materials with perovskite crystal structure; High piezoelectric effect and low dielectric loss; Simple fabrication process, compatible with MEMS fabrication; Highly toxic due to the presence of lead.	Most are modified or doped PZT, such as Lead Magnesium Niobate-PZT (PMN-PZT), PZT-5A, Zinc Oxide-enhanced PZT (PZT-ZnO), etc.
Lead-free Piezoceramics	Non-toxic piezoceramics; Have lower transduction efficiency; Competitive lead-free materials are perovskite crystal structured type.	BaTiO3; Bismuth Sodium Titanate (BNT-BKT); Potassium Sodium Niobate (KNN)-based material: LS45, KNLNTS.
Piezopolymers	Electroactive Polymer (EAP); Flexible, non-toxic, and light-weight; Smaller electromechanical coupling than piezoceramics; Low manufacturing cost and rapid processing; Biocompatible, biodegradable, and low power consumption compared to other piezoelectric materials.	Can be used for piezo-MEMS fabrication; Polyvinylidene Fluoride (PVDF)-derived polymers.

* Reproduced with permission from from [5]. Copyright © 2023 Elsevier B.V.

**Table 3 sensors-23-05841-t003:** Piezoelectric materials and their properties *.

Material Properties	Symbol	ZnO	PMN-32%PT	PZT-4	PZT-5A	PZT-5H	BaTiO3	PVDF
**Relative dielectric** **constant (1 kHz)**	$K_{33}^{T}$	8.67	1620	1475	1600		1436	13.5
	$K_{31}^{T}$	11.26	7000	1300	1800	3800	1680	
**Piezoelectric charge** **constant (10^−12^ C/N)**	$d_{31}$	−5.12	−760	−123	−190	−320	−79	25
	$d_{33}$	12.3	1620	289	390	650	191	−23
	$d_{15}$	−8.3	192	496	460	1000	270	
**Piezoelectric voltage** **constant (10^−3^ Vm/N)**	$g_{31}$	−0.45	−12.29	−11.1	−11.3	−9.5	−4.7	210
	$g_{33}$	1.09	26.15	26.1	23.2	19	11.4	−330
	$g_{15}$		13.39	39.4	32.4	35.5	18.8	
**Electromechanical** **coupling coefficients**	$k_{31}$	0.18	0.44	0.33	0.35	0.44	0.21	0.1
	$k_{33}$	0.47	0.93	0.7	0.72	0.75	0.49	
	$k_{t}$	0.23	0.62	0.58	0.49	0.55		0.12
**Energy conversion** **efficiency**	$k_{31}^{2}$	0.03	0.19	0.11	0.12	0.19	0.04	0.01
	$k_{33}^{2}$	0.22	0.86	0.49	0.52	0.56	0.24	
	$k_{t}^{2}$	0.05	0.38	0.34	0.24	0.30		0.01
**Mechanical quality factor**	$Q_{m}$		69	500	80	32	300	3~10
**Dielectric loss**	tan⁡δ		0.42%	0.4%	0.02%	2%		
**Curie temperature (°C)**	$T_{C}$	554	145	328	350	225	115	100
**Operation frequency**	N/A	Up to GHz	Up to GHz	Up to GHz	Up to GHz	Up to GHz	Up to GHz	Up to MHz
**Minimum size**	N/A	Down to nm	Down to nm	Down to nm	Down to nm	Down to nm	Down to nm	Down to nm

* Data from Yang et al. [8], Piezo.com [56], CTS Corporation [57], Uchino [58], Kobiakov [59], Berlincourt et al. [60], PolyK Technologies [61].

**Table 4 sensors-23-05841-t004:** Mean (±standard deviation) plantar pressure (Kpa) in the 10 regions *.

Area	Martínez-Nova	Putti	Fernández-Seguín	Xu	Bryant
Heel	253.1 ± 20.2	264.3 ± 44.1	270.13 ± 6.15	237.9 ± 50.1	167 ± 24
Midfoot	65.9 ± 16.8	109.0 ± 38.5	28.62 ± 1.48	65.3 ± 27.3	39 ± 25
Met Head 1	308.2 ± 36.1	248.0 ± 70.1	55.56 ± 3.53	178.3 ± 38.3	122 ± 33
Met Head 2	405.8 ± 57.4	246.5 ± 48.3	123.03 ± 4.86	367.5 ± 87.9	188 ± 41
Met Head 3	394.1 ± 37.7	224.7 ± 50.4	157.44 ± 3.06	344.6 ± 101.4	154 ± 32
Met Head 4	203.6 ± 22.5	161.0 ± 49.7	114.98 ± 3.22	234.6 ± 56.3	114 ± 39
Met Head 5	118.4 ± 18.3	141.6 ± 58.4	52.89 ± 2.66	116.4 ± 31.2	89 ± 43
Hallux	146.5 ± 22.5	280.4 ± 83.0	100.14 ± 3.46	161.6 ± 48.9	139 ± 38
Lesser Toes	105.3 ± 14.3	130.3 ± 55.3	27.51 ± 2.41	47.1 ± 22.3	83 ± 25

* Data from Martínez-Nova [79], Putti [80], Fernández-Seguín [81], Xu [82], Bryant [83].

**Table 5 sensors-23-05841-t005:** Performance of four types of PFEHs.

Structure	Reference	Material	Size (mm × mm × mm)	Average Power	Location
Flat plate	Kymissis J et al. [120]	PVDF	100 × 80 × 2.45	1.0 mW@1 Hz	On the sole
	Jeong SY et al. [17]	PZT	60 × 40 × 7	0.8 mW@1 Hz	On the sole
	Ahmad N et al. [19]	PZT	Ø27, thick 0.6, total 5 discs	1.41 mW@1 Hz	On insole
	Chaudhary P et al. [129]	PZT	A Ø50 disc	0.2 mW@6 km/h	Inside sole
Curved	Kymissis J et al. [120]	PZT	70 × 70 × 7	2.0 mW@1 Hz	On the sole
	J Zhao et al. [106]	PVDF	80 × 50 × 0.24	1.0 mW@1 Hz	On the sole
	Jung WS et al. [105]	PVDF	70 × 40 × 0.6	0.5 mW@0.5 Hz	Inside insole
Cantilever	L Moro et al. [13]	PZT	20 × 14 × 0.4	13.8 μW@1 Hz	Heel
	Xin Y et al. [130]	PVDF		495.8 μW	Heel
	Fan et al. [137]	PZT	19.1 × 7.1 × 0.245	0.35 mW@8 km/h	Heel
Flextensional	Leinonen et al. [16]	PZT	Ø35, thick 2.7	0.8 mW@1 Hz	Heel
	Y Kuang et al. [112]	PZT	52 × 30 × 16.2	2.5 mW@4.8 km/h	Inside sole
	Qian F et al. [14]	PZT	94 × 68 × 24	9.3 mW@4.8 km/h	Heel
	Qian F et al. [69]	PZT		23.9 mW@2 Hz	Heel

## Data Availability

No new data were created or analyzed in this study. Data sharing is not applicable to this article.

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
