# Peer review of "A Review of Piezoelectric Footwear Energy Harvesters: Principles, Methods, and Applications"

_sensors, 2023, doi:10.3390/s23135841_

Round 1
Reviewer 1 Report
The work is well-organized and nicely presented. However the work is covering a very narrow area (footwear energy harvesting) which doesn't justify a review article. At the same time footwear energy harvesting isn't a very popular approach in energy harvesting domain due to several reasons. Hence, the article is less likely will be impactful for the community.
The article cited a lot of papers. Most/many of those are not aligned with the focus of this article. For example, a lot of papers were cited in section-2 and -3 where the article is covering some basic/background informations. Much more articles are cited only in these two sections than footwear energy harvesting methods. A lot of self-citation.
Author Response
Thank you very much for your time and efforts to review and provide the feedback. We appreciate your concerns about the narrow focus of the article and its potential impact on the community. We understand that footwear energy harvesting is a sub area of piezoelectric energy harvesting. The total number of journal papers is only in the hundred level, which is significantly smaller than entire piezoelectric energy harvesting with total journal papers up to 10,000. However, we believe that our review provides a valuable contribution by consolidating the current knowledge in this emerging field, because i) this a one of the critical area to provide renewable power source for a) healthcare sensor nodes and monitoring system, b) portable electronic equipment and communication tools for military mentions; 2) you might see will provide interdisciplinary insides to readers on the entire off-resonance piezoelectric energy harvesting through this review platform.
We apologize if it seemed that we cited papers not directly aligned with the focus of the article. However, if you see the reference content directly, most of the papers are related to footwear energy harvester. We carefully revise the citations to ensure better alignment. If you have a chance to press “Ctrl+A” in the word document, it is easily to see the citations are evenly distributed in the whole paper like the screen shoot below. We understand that reviewers will take much more time to review a review paper than a regular research paper.
Regarding self-citations, we assure you that our intention was not to inflate our own contributions but to reference relevant research we have conducted. We evaluated the necessity of these self-citations and revise accordingly. We took out two of our papers[1, 2], which emphasize the imports to use “33” mode piezoelectric structure for piezoelectric device either acting as energy harvester or actuator to increase energy efficiency.
References
- Xu, T.-B., et al., A single crystal lead magnesium niobate-lead titanate multilayer-stacked cryogenic flextensional actuator. Applied Physics Letters, 2013. 102(4).
- Tolliver, L., T.-B. Xu, and X. Jiang, Finite element analysis of the piezoelectric stacked-HYBATS transducer. Smart Materials and Structures, 2013. 22(3).

Reviewer 2 Report
The manuscript was interesting and well-return. The following list of comments will help to further improve the manuscript:
1. I suggest that the author include additional information in Table 2 to make it more helpful to the reader. Specifically, the table could benefit from the inclusion of minimum operating frequency, efficiency, and minimum size of the piezoelectric material. This would provide more comprehensive information and assist the reader in making informed decisions regarding the use and selection of piezoelectric materials.
2. Figure 20a, and figure 20b images can be improved and enlarged
3. Author can compare other mode of energy like rotary magneto drive and state there advantage and disadvantage over Piezoelectric harvesting system
4. Also, I request author to provide future scope with other possibilities of using piezoelectric harvesting system, since, one can harvest maximum of 23.9mv@ Hz
5. In Table 4, it will be appropriate to add size of the PZT material used.
Author Response
Authors response to reviewer-2 comments for Sensor 2329913
Review-2 (3/30/2023)
Comments and Suggestions for Authors
The manuscript was interesting and well-return. The following list of comments will help to further improve the manuscript:
- I suggest that the author include additional information in Table 2 to make it more helpful to the reader. Specifically, the table could benefit from the inclusion of minimum operating frequency, efficiency, and minimum size of the piezoelectric material. This would provide more comprehensive information and assist the reader in making informed decisions regarding the use and selection of piezoelectric materials.
The additional information, such as Electromechanical coupling coefficients, Energy conversion efficiency, Operation frequency, and Minimum size, have been added in the revised Table 2 (Now Table 3).
- Figure 20a, and figure 20b images can be improved and enlarged
Figure 20a, and figure 20b were improved.
- Author can compare other mode of energy like rotary magneto drive and state their advantage and disadvantage over Piezoelectric harvesting system
The comparisons of advantages and disadvantages for piezoelectric, electromagnetic, and triboelectric energy harvesting methods are Added in the new Table 1.
- Also, I request author to provide future scope with other possibilities of using piezoelectric harvesting system, since, one can harvest maximum of 23.9mv@ Hz
This is a very interest question. The Two-stage force amplification piezoelectric transducer/harvester, which was used achieving 23.9mv, was referred the invention [1]. We have a couple of manuscripts related to the knowledge and potential applications. However, due to the intellectual properties and interests of Dr. Xu’s industry partners on several NASA and Airforce projects. The papers are on hold to release. Dr. Xu recently discovered that NASA released one of his award receipt speech slide[2] on the internet. One of the slides indicated that this energy harvester has at lease one order of magnitude higher power density than others either working as resonance mode or off-resonance mode. For instance, the Comparison With the State-of-the-Art Piezoelectric Energy Harvesters in the following tables, which cited in that slide.
Table 1. Comparison with the State-of-the-Art Piezoelectric Energy Harvesters at off-resonance mode operation
Table 2. Comparison with the State-of-the-Art Piezoelectric Energy Harvesters at resonance mode operation
- In Table 4, it will be appropriate to add size of the PZT material used.
The size of each piezo material used are added in the revised Table 4 (Now Table 5).
- Tian-Bing Xu, E.J.S., Lei Zuo, Xiaoning Jiang, Jin Ho Kang, Multistage force amplification of piezoelectric stacks. 2015: US.
- Tian-Bing Xu, J.H.K., Emilie J. Siochi, Lei Zuo, Wanlu Zhou, and Xiaoning Jiang, Ultra-High Power Density Piezoelectric Energy Harvesters. 2015, Energy Harvesting and Storage USA International Conference and Exhibition 2015.
